# The Inverted U-Shaped Association between Serum Vitamin D and Serum Uric Acid Status in Children and Adolescents: A Large Cross-Sectional and Longitudinal Analysis

**DOI:** 10.3390/nu16101492

**Published:** 2024-05-15

**Authors:** Zhuang Ma, Ting Xiong, Yan Li, Binxuan Kong, Wenlong Lu, Ziyang Zhang, Liangkai Chen, Yuhan Tang, Ping Yao, Jingfan Xiong, Yanyan Li, Yuanjue Wu

**Affiliations:** 1Department of Nutrition and Food Hygiene, School of Public Health, Guangzhou Medical University, Guangzhou 511436, China; 2020115092@stu.gzhmu.edu.cn (Z.M.); xiongkoala@gzhmu.edu.cn (T.X.); 2Shenzhen Center for Chronic Disease Control, Shenzhen 518020, China; nyan0727@outlook.com (Y.L.); lulu990461@126.com (W.L.); 15112361106@163.com (Z.Z.); xiongjingfan@126.com (J.X.); 3Department of Nutrition and Food Hygiene, Hubei Key Laboratory of Food Nutrition and Safety and the Ministry of Education (MOE) Key Laboratory of Environment and Health, School of Public Health, Tongji Medical College, Huazhong University of Science and Technology, Wuhan 430030, China; apricotkong@163.com (B.K.); clk@hust.edu.cn (L.C.); 2015220157@hust.edu.cn (Y.T.); yaoping@mails.tjmu.edu.cn (P.Y.)

**Keywords:** serum vitamin D, serum uric acid, children and adolescents, cross-sectional, longitudinal surveys

## Abstract

Background: Serum vitamin D is associated with hyperuricemia. However, previous studies have been controversial, with limited focus on children and adolescents. Objective: This study aimed to examine the cross-sectional and longitudinal associations between serum vitamin D and serum uric acid (SUA) levels in children and adolescents. Methods: The cross-sectional survey comprised 4777 participants aged 6 to 18 years, while the longitudinal survey involved 1641 participants aged 6 to 12 years, all derived from an ongoing cohort study in Shenzhen, China. Restricted cubic splines were used to visualize the dose–response relationship between vitamin D and SUA and the risk of higher SUA status. Two-segment generalized linear models (GLM) and logistic models were used to assess the association between vitamin D and SUA and higher SUA status, respectively. The longitudinal analysis used GLM. Results: We observed an inverted U-shaped relationship between vitamin D and SUA (*p*-overall < 0.0001, *p*-nonlinear = 0.0002), as well as the risk of higher SUA status (*p*-overall = 0.0054, *p*-nonlinear = 0.0015), with the vitamin D inflection point at 24.31 and 21.29 ng/mL, respectively. A 10 ng/mL increment in 25(OH)D_3_ levels, when below 20.92 ng/mL, was associated with a 68% rise in the risk of higher SUA status (OR: 1.68, 95%CI: 1.07–2.66). Conversely, when 25(OH)D_3_ levels were above or equal to 20.92 ng/mL, a 10 ng/mL increment was associated with a 45% reduction risk of higher SUA status (OR: 0.55, 95%CI: 0.36–0.84). Longitudinal analysis indicated that the annual change of SUA was from −4.80 (β, 95%CI: −10.74, 1.13) to −9.00 (β, 95%CI: −15.03, −2.99) and then to −6.77 (β, 95%CI: −12.83, −0.71, *p* for trend = 0.0212) μmol/L when increasing the quartile of vitamin D_3_. Conclusions: An inverse U-shaped relationship was observed between vitamin D and SUA as well as the risk of higher SUA status. Sufficient vitamin D levels appear to play a preventative role against the age-related increase in SUA. Ensuring adequate vitamin D levels may be beneficial in improving uric acid metabolism.

## 1. Introduction

Serum uric acid (SUA), a metabolite of purine catabolism, serves as a vital antioxidant in the human body, effectively scavenging harmful free radicals [1]. However, abnormally increasing SUA levels can lead to gout and various comorbidities, including metabolic syndrome, hypertension, and diabetes, posing significant health risks [2,3,4,5]. Alarmingly, the prevalence of hyperuricemia has increased significantly worldwide over recent years [6], particularly high among children and adolescents. Research data from China, the U.S., and some European countries have indicated an estimated overall prevalence varies from 0.6% to 50.4% among children and adolescents [7,8,9].

Vitamin D, a fat-soluble vitamin, encompasses two primary forms: vitamin D_2_, predominantly originating from ergosterol-rich, plant-based foods (yeast and mushrooms) [10], and vitamin D_3_, mainly sourced from animal-derived foods or photochemical reactions in the skin. Vitamin D, essential for bone health and calcium homeostasis, has emerged as a focal point due to its extensive role beyond skeletal functions, influencing various metabolic pathways and disease susceptibilities [11,12,13,14,15]. UA, an inert waste product of body metabolism, is associated with various metabolic disorders [16]. Previous studies have demonstrated that vitamin D may indirectly influence SUA concentrations. Vitamin D deficiency activates the parathyroid gland [17], leading to the release of parathyroid hormone [18]. Simultaneously, an increasing body of literature indicates that parathyroid hormone (PTH) impacts the excretion of UA. However, vitamin D deficiency is widespread in populations worldwide [19], especially among children and adolescents. Various studies conducted in the U.S., Europe, and China have highlighted the prevalence of vitamin D deficiency in children and adolescents, ranging from a concerning 4% to a startling 40% [20,21,22,23]. Given the high prevalence of hyperuricemia and vitamin D deficiency in children and adolescents, it is of great significance to explore the association between vitamin D and SUA.

However, the relationship between serum vitamin D levels and SUA in children and adolescents remains poorly understood, with only a few conflicting observational studies in adults available. Some studies suggest a link between vitamin D deficiency and elevated risk of hyperuricemia in adults [24,25], whereas others report contradictory findings [26,27]. Notably, children and adolescents are in critical growth and developmental stages, during which their bodies exhibit high metabolism and strong volatility due to physiological demands. Puberty triggers a significant increase in hormone levels, markedly affecting various metabolic and absorptive pathways [28]. Consequently, the dynamics of vitamin D absorption and metabolism, along with UA production and metabolism, may significantly diverge in children and adolescents when compared to adults [29]. This suggests that the association between serum vitamin D and SUA in children and adolescents could significantly differ from that in adults.

Despite the potential importance of this relationship, prior investigations into the connection between vitamin D and SUA have largely overlooked children and adolescents. This study seeks to fill this gap through the implementation of both cross-sectional and longitudinal analyses.

## 2. Materials and Methods

### 2.1. Participants

This study conducted an analysis using data from Evaluation and Monitoring on School-based Nutrition and Growth in Shenzhen (EMSNGS), an ongoing cohort of children and adolescents established and managed by the Shenzhen Center for Chronic Disease Control, which was initiated in 2021 (Chinese clinical trial registry: ChiCTR2100051722; https://www.chictr.org.cn (accessed 20 December 2023). The primary aim of this cohort is to evaluate and monitor the nutritional health, growth, and development of children and adolescents aged 6 to 18 years in Shenzhen. The sample collection for this cohort utilized a multistage, stratified cluster, random sampling approach [30].

All participants and their legal guardians were fully informed about the details of the cohort protocol and aims. Before participation, written informed consent was obtained from all participants or their legal guardians to ensure ethical compliance and participant autonomy. Furthermore, the study protocol underwent rigorous review and approval by the ethics review committee of the Center for Chronic Disease Control (no. SZCCC-2021-037-01-PJ, 25 April 2021).

The study initially involved a total of 5348 participants. Among them, 195 participants were excluded due to their disagreement with the study protocol, and 35 participants were excluded due to the absence of essential information. Participants who fell outside the age range of 6 to 18 years were then excluded (*n* = 18). Individuals without data on 25-hydroxyvitamin D [25(OH)D] or SUA were also excluded from the study (*n* = 154). Furthermore, exclusion criteria encompassed participants with birth defects, congenital developmental abnormalities, endocrine disorders (such as growth hormone deficiency, thyroid dysfunction, and diabetes), autoimmune disorders, hematologic or neoplastic disorders, and chronic liver and kidney disease (*n* = 125) in addition to those lacking information on these conditions (*n* = 44). After applying the exclusion criteria, a total of 4777 participants were enrolled in the cross-sectional analysis (Appendix A). Our longitudinal analysis, including 1793 individuals, was conducted on students in grades 1–5 from the first round of follow-up in this cohort. Of these, 152 participants without SUA data were excluded, leaving a total of 1641 individuals enrolled in the subsequent analyses.

### 2.2. Assessment of 25(OH)D, 25(OH)D_2_, and 25(OH)D_3_

Serum 25(OH)D levels are the sum of serum 25(OH)D_2_ and 25(OH)D_3_. The quantification of serum 25(OH)D_2_ and 25(OH)D_3_ levels was determined through liquid chromatography–mass spectrometry (LC-MS, Agilent 6470, Agilent, Santa Clara, CA, USA), exhibiting a precision exceeding 90%. The suitable serum samples for direct examination were procured following a meticulous protocol, and the serum samples were transported to the biobank of the Center for Chronic Disease Control for further analysis [30]. All serum samples were meticulously stored in a purpose-built −80 °C refrigerator before the measurement, thereby guaranteeing their utmost integrity and stability.

### 2.3. Assessment of SUA and Higher SUA Status

SUA concentrations were measured using colorimetry on an automated analyzer (AU5811, Beckman Coulter Inc., Brea, CA, USA) at the laboratory of the Center for Chronic Disease Control. Given the absence of a well-established reference cut-off value for hyperuricemia among children and adolescents, we assigned a higher SUA status and established diagnostic criteria based on discrimination within both gender and age groups. Specifically, we categorized the subjects into two distinct age subgroups: 6 to 12 and 12 to 18 years. SUA levels exceeding the 90th percentile of the overall distribution within the respective gender and age subgroups were considered indicative of a higher SUA status.

### 2.4. Other Covariates

The socio-demographic and economic characteristics, lifestyle, and health-related data of the participants were obtained through rigorous questionnaire interviews, which were administered by trained professionals. Furthermore, covariates that necessitated manual or instrumental measurement, including height, waist, and weight, were accurately measured by professional staff employing standardized methodologies. Weight status was determined by categorizing individuals as underweight, normal, overweight, or obese based on age- and sex-specific body mass index (BMI, calculated as weight divided by the square of height) cut-off points established by the screening criteria for nutritional status of Chinese school-aged children and adolescents [31,32]. The specific evaluation criteria for weight status were determined as follows: Overweight was defined as having a BMI equal to or greater than the 85th percentile (P85) but less than the 95th percentile (P95) of the corresponding sex and age group; obesity was characterized by a BMI equal to or greater than the 95th percentile (P95) of the corresponding sex and age group; and underweight was identified as a BMI equal to or less than the 5th percentile (P5) of the corresponding sex and age group. Pubertal development stages were evaluated by a pediatrician specialized in endocrinology using the Tanner staging system, and participants were further categorized into three categories: pre-puberty (Tanner stage I), mid-puberty (Tanner stages II–III), and post-puberty (Tanner stage IV and above) [33,34,35]. Household income was categorized into three groups: <CNY 120,000 (Chinese Yuan) (CNY 1 ≈ EUR 0.13; CNY 1 ≈ USD 0.14), CNY120,000 to CNY 250,000, and ≥CNY 250,000. The education levels of parents were classified into four categories: ≤9 years, 9–12 years, 12–15 years, and >15 years. Regarding smoking status, participants were classified into four groups: never smoked, <1 cigarette/month, ≥1 cigarette/month, and e-cigarette usage. The drinking status was classified as never, <1 standard drink/month, and ≥1 standard drink/month. Classifications of the levels of moderate-to-vigorous physical activity (MVPA) were less than 0.5 h/day, 0.5–0.9 h/day, 1.0–2.9 h/day, and ≥3 h/day. Central obesity was defined as a waist circumference at or above the 90th percentile for age and sex [36].

### 2.5. Statistical Analyses

Before calculating the baseline characteristics of the participants, continuous variables were assessed for normality using the Kolmogorov–Smirnov test. Subsequently, continuous variables with normal distributions were reported as means with standard deviations (SD), while non-normal continuous variables were reported as medians along with their interquartile ranges (IQR). Categorical variables were described as frequencies and percentages. We employed analysis of variance (ANOVA) for continuous variables with normal distributions, the Kruskal–Wallis rank test for continuous variables with non-normal distributions, and the chi-square test for categorical variables to compare the serum 25(OH)D groups.

We performed a log-transformation for 25(OH)D_2_ to eliminate its severely skewed distribution. Restricted cubic splines (RCS) with three knots were used to describe the dose–response relationships between serum 25(OH)D, log-25(OH)D_2_, 25(OH)D_3_ levels, and SUA levels. Furthermore, a threshold effect analysis was conducted leveraging a two-segment generalized linear regression model to detect inflection points in the inverted U-shaped, nonlinear relationship, and the results were expressed as coefficients (β) with 95% confidence intervals (CIs). Two regression models were employed to control for confounding factors. Model I accounted for age and sex. Model II further incorporated adjustments for weight status, pubertal development stages, household income, maternal education levels, smoking and drinking status, vitamin D supplement, multivitamin/mineral supplement, and levels of MVPA. Subsequently, we conducted stratified analyses to investigate the threshold effect of vitamin D on SUA across various subgroups, and the interactions were performed by Wald test. These subgroups included ages (6 to 12 or 12 to 18 years), gender, stage of puberty (pre-, mid-, or post-puberty), and weight status (underweight, normal, or overweight/obesity) and central obesity (yes or no). Similarly, we used RCS with three knots to describe the dose–response relationship between serum 25(OH)D, Log-25(OH)D_2_, and 25(OH)D_3_ levels and the risk of higher SUA status. We further employed threshold effect analysis based on the dichotomous logistic regression model to detect the inflection points of the inverted U-shaped relationship in them, and the results were exhibited as odds ratios (ORs) and 95% confidence intervals (CIs).

Following the subtraction of baseline SUA data from the corresponding SUA follow-up data, the resulting difference was divided by the duration of follow-up (years) to determine the annual change in SUA levels. Generalized linear models adjusting the same confounding factors were employed to investigate the association between quartiles of serum 25(OH)D, 25(OH)D_2_, 25(OH)D_3_, and the annual changes in SUA levels. Furthermore, to evaluate the significance of linear trends within the quartiles of serum 25(OH)D (either 25(OH)D_2_ or 25(OH)D_3_), the median value of each quartile was assigned and subsequently treated as a continuous variable in the statistical models.

The statistical analysis was conducted using R software version 4.3.1 (R Foundation for Statistical Computing, Vienna, Austria). A significance level of *p* < 0.05, with two-sided testing, was employed to determine statistical significance.

## 3. Results

### 3.1. Baseline Characteristics

The present study enrolled a total of 4777 children and adolescents, with an overall median age of 11.9 years (IQR: 9.2–14.7), and 44.5% were female. Among the participants, 31.3% were categorized as pre-puberty, 22.2% as mid-puberty, and 46.5% as post-puberty. Table 1 presents the demographic characteristics of the participants categorized by quartiles of serum 25(OH)D levels. Compared to the lowest quartile (Q1), the high vitamin D group was characterized by a younger age, non-alcohol consumption, a higher proportion of men, and a smaller waist circumference. The Q4 group exhibited a higher prevalence of being underweight and overweight, a greater likelihood of being in the pre-puberty stage, higher household income levels, and greater inclination to take vitamin D supplements. Conversely, the Q1 group showed a greater probability of engaging in lower levels of MVPA compared to the other groups (Table 1). Appendix A presents the subject characteristics of the longitudinal analysis. Among the 1641 included subjects, the median age was 10.33 (IQR: 9.04, 11.53), with 45% being female. Compared to those in the lowest quartile (Q1) of 25(OH)D, participants in higher quartiles tended to be younger, female, prepubertal, with higher household income and higher parental education levels, and greater likelihood of receiving vitamin D supplementation. Furthermore, an annual change in SUA exceeding 0 indicated that participants experienced an elevation in SUA levels during follow-up compared to their baseline serum uric acid levels. This observation aligns with prior findings indicating that SUA tends to increase with age in children and adolescents [37]. As the vitamin D level increased, the annual variation of SUA tended to decrease (Appendix A).

### 3.2. Association between Serum 25(OH)D, Log-25(OH)D_2_, 25(OH)D_3_, and SUA Levels

The RCS graphically demonstrated the inverted U-shaped, nonlinear dose–response relationship between serum 25(OH)D (*p*-overall < 0.0001, *p*-nonlinear = 0.0002; Figure 1), log-25(OH)D_2_ (*p*-overall = 0.0066, *p*-nonlinear = 0.0034), 25(OH)D_3_ (*p*-overall < 0.0001, *p*-nonlinear < 0.0001), and SUA levels. In the comprehensive model (model II), further threshold effect analysis revealed that when serum 25(OH)D levels were below 24.31 ng/mL, each 10 ng/mL increment in 25(OH)D was associated with an elevation of SUA levels by 9.15 (95% CI: 1.48, 16.83, *p* = 0.019; Table 2) μmol/L. Similarly, for 25(OH)D_3_, a 10 ng/mL increment below the threshold of 23.79 ng/mL was associated with an SUA increase of 12.15 (95% CI: 4.56, 19.74, *p* = 0.008) μmol/L. Each unit decrease in log-25(OH)D_2_ (corresponds to a 10-fold increase in 25(OH)D_2_) was associated with a change of SUA levels by −16.80 (95%CI: −29.22, −4.39, *p* = 0.008) μmol/L when exceeding −0.389 ng/mL. Interestingly, we observed a trend of negative association with SUA in the second half of the inflection points for 25(OH)D and 25(OH)D_3_, but significance was not found. Furthermore, a positive trend with SUA in the first half of the inflection point of log-25(OH)D_2_ was observed despite non-significance.

### 3.3. Stratified Analyses of the Association between Serum 25(OH)D and SUA Levels

In the stratified analysis of the threshold effect in the relationship between serum 25(OH)D and SUA, we observed a significant interaction between serum 25(OH)D levels and the participant’s sex (male or female) (*p* for interaction < 0.0001; Appendix A), age (6 to 12 or 12 to 18 years) (*p* for interaction = 0.0008), and pubertal stage (pre-, mid-, or post-puberty) (*p* for interaction = 0.0332) in the pre-inflection phase. This indicates that the intricate association between serum 25(OH)D and SUA levels exhibited variations across different sexes, ages, and pubertal stages. Furthermore, we did not observe or detect any interaction of serum 25(OH)D with weight stages (underweight, normal, or overweight/obesity), central obesity (yes or no), and any subgroup differences beyond the inflection point (all *p* for interaction > 0.05).

### 3.4. Associations between Serum 25(OH)D, Log-25(OH)D_2_, 25(OH)D_3_, and Risk of Higher SUA Status

The RCS visually exhibited the inverted U-shaped, nonlinear dose–response relationship between serum 25(OH)D (*p*-overall = 0.0054, *p*-nonlinear = 0.0015; Figure 2), log-25(OH)D_2_ (*p*-overall = 0.1188, *p*-nonlinear = 0.0398), 25(OH)D_3_ (*p*-overall = 0.0028, *p*-nonlinear = 0.0007), and risks of higher SUA status. Results from further two-segment logistic regression analysis in model II revealed that a 10 ng/mL increment in the 25(OH)D was associated with a 52% (95%CI: 0.92–2.31, *p* = 0.11) elevation in the risk of having a higher SUA status when the serum 25(OH)D level was below 21.29 ng/mL (Table 3) and a 10 ng/mL decrement in the 25(OH)D was associated with a 48% (95%CI: 0.34–0.78, *p* = 0.002) reduction in the risk of high SUA status when the serum 25(OH)D level was greater than or equal to 21.29 ng/mL. Furthermore, the inflection point for serum 25(OH)D_3_ was 20.92 ng/mL. Each 10 ng/mL increment in 25(OH)D_3_ was associated with a 68% (95%CI: 1.07–2.66, *p* = 0.025) rise in the risk of higher SUA status before inflection point, and each 10 ng/mL decrease in 25(OH)D_3_ associated with a 45% (95%CI: 0.36–0.84, *p* = 0.006) reduction in the risk of higher SUA status beyond inflection point.

### 3.5. Longitudinal Analysis of the Association between Serum 25(OH)D, 25(OH)D_2_, 25(OH)D_3_, and SUA Levels

A total of 1641 participants were followed-up, with a median duration of 1.56 (IQR: 1.52–1.60) years. The longitudinal analysis revealed that elevated levels of baseline serum 25(OH)D were associated with a lower annual increase in SUA (Figure 3, Table 4). Compared with participants of the Q1 group of serum 25(OH)D levels, the β values (represents the annual increase in SUA) for successive quartiles were −3.91 (95%CI: −9.84, 2.02) to −9.27 (95%CI: −15.30, −3.24) and then to −6.35 (95%CI: −12.40, −0.30, *p* for trend = 0.0225) μmol/L. A similar trend for the quartiles of 25(OH)D_3_ was also observed, ranging from −4.80 (95%CI: −10.74, 1.13) to −9.00 (95%CI: −15.03, −2.99) and then to −6.77 (95%CI: −12.83, −0.71, *p* for trend = 0.0212) μmol/L. However, 25(OH)D_2_ merely exhibited a trend and did not have statistical significance, ranging from −4.34 (95%CI: −10.19, 1.51) to –1.96 (95%CI: −7.90, 3.98) and then to −5.91 (95%CI: −11.83, 0, *p* for trend = 0.0929) μmol/L.

## 4. Discussion

In our cross-sectional analysis, we observed an inverted U-shaped, nonlinear relationship between serum 25(OH)D, log-25(OH)D_2_, 25(OH)D_3_, and SUA levels with the inflection points of 24.31, −0.389, and 23.79 ng/mL, respectively. Furthermore, our investigation revealed an inverted U-shaped, nonlinear association between serum 25(OH)D, log-25(OH)D_2_, 25(OH)D_3_, and the risk of higher SUA status, with cut-off values at 21.29, −0.356, and 20.92 ng/mL, respectively. Finally, we observed that higher serum 25(OH)D (25(OH)D_3_) levels were positively associated with a lower annual SUA rise in longitudinal analyses.

Previous studies have published different results about the relationship between serum vitamin D and SUA in adults. A cross-sectional study about elderly men conducted by Seibel demonstrated a positive association between SUA and 25(OH)D (β = 0.09, *p* = 0.005) [26]. A cohort of patients with chronic kidney diseases (CKD) and type 2 diabetes from another multicenter clinical study about hepatitis patients revealed a similar positive correlation [27,38]. However, most current studies tend to favor a negative correlation between 25(OH)D and SUA. For instance, a study in Chinese postmenopausal women indicated that insufficient vitamin D is significantly associated with elevated SUA (Q1 vs. Q4, OR, 2.38, *p* = 0.001) [39]. Another study of 84 vitamin D-deficient elderly Egyptians found an inverse association between vitamin D and SUA levels (r = −0.924, *p* = 0.0001) [40]. Furthermore, a cross-sectional survey conducted among U.S. adults revealed a tendency towards an inverse correlation between serum vitamin D and hyperuricemia (Q1 vs. Q4, HR: 1.46) [24]. Similarly, another cross-sectional study involving 15,723 U.S. adults demonstrated that individuals with higher serum vitamin D levels exhibited a lower risk of hyperuricemia [41]. In summary, previous studies remain controversial and insufficient, which may be attributed to population and geographic differences and sample-size limitations. Consequently, the relationship between serum vitamin D and SUA needs to be further discussed in wider population and longitudinal studies.

To our knowledge, our study pioneers the exploration of the relationship between serum vitamin D and SUA levels in children and adolescents. We proposed a postulation in our cross-sectional survey: the existence of an inverted U-shaped correlation between these two factors. Although the second half of the survey failed to attain statistical significance, a trend toward a negative association was also observed. We speculated that this may have been due to the limited sample size, resulting in insufficient statistical power. Given that there was a lack of standards for the diagnosis of hyperuricemia in children and adolescents, we defined high SUA status as SUA levels surpassing the 90th percentile for age and sex. The strategy allowed us to focus on the extreme SUA range, not only re-examining the relationship between SUA and serum vitamin D but also facilitating the identification of high-risk groups predisposed to hyperuricemia. Crucially, our findings corroborate our earlier hypothesis that a re-increase in serum vitamin D level beyond the inflection point significantly reduces the risk of elevated SUA status. Furthermore, our longitudinal analysis once again reinforces our hypothesis that an increase in serum vitamin D level has a positive effect on inhibiting SUA rise with age. Unlike previous studies, our research focused on the longitudinal association of vitamin D and SUA while proposing the cross-sectional hypothesis, indicating that higher levels of vitamin D are associated with a lower rise in the annual increase of SUA.

The interactive mechanism underlying the relationship between serum vitamin D and SUA levels remains elusive. Based on our findings, it is reasonable to speculate that there are differences in the produced and metabolic process of SUA under varying vitamin D concentrations. Notably, a previous study reported the cut-off point for vitamin D insufficiency in children and adolescents was around 20 ng/mL [42], which was close to our inflection point. Therefore, before reaching the inflection point, vitamin D exists in a state of insufficiency, and as its levels rise, it prioritizes maintaining critical physiological functions such as regulating calcium and phosphorus homeostasis, promoting calcium absorption, and safeguarding bone health [11]. During this process, renal reabsorption of calcium increases. The excretion of uric acid, which primarily relies on the uric acid transporter 1 located on renal tubular cell membranes, competes with this process, potentially impeding uric acid excretion. Furthermore, the loss of uricase activity during renal retention may exacerbate SUA accumulation [43]. Consistent with our inference, prior investigations exploring the link between total calcium and SUA in American adolescents have demonstrated a positive linear correlation between them [44]. However, once vitamin D surpasses the threshold, it begins to exert a broader range of physiological functions. Elevated vitamin D levels can suppress parathyroid activity, leading to a reduction in serum PTH concentration [17], which in turn can modulate the secretion and transport of uric acid [18]. Furthermore, higher vitamin D concentrations positively impact insulin sensitivity [45], and prior research has demonstrated an inverse association between insulin resistance and renal clearance of SUA [46]. Additionally, vitamin D exerts its influence on numerous physiological processes by targeting various genes [47]. It is conceivable that vitamin D may regulate the expression of uric acid-associated genes, such as SLC2A9 and SLC17A3, among others, and then affect the production of uric acid.

In our stratified analysis, we observed significant differences in the association between vitamin D and SUA levels among different genders, ages, and pubertal stage subgroups. These findings align with prior studies that have highlighted sex- and age-related disparities in vitamin D status and uric acid concentrations. Furthermore, estrogen plays a role in regulating both vitamin D and SUA metabolism [48,49], which may contribute to the observed differences. However, these differences need to be further validated in a broader population and through longitudinal studies.

The primary strength of our current study lies in the large and representative sample that was obtained through stratified cluster random sampling. This approach ensured robust statistical power and allowed us to draw meaningful conclusions. Furthermore, by integrating cross-sectional and longitudinal surveys, our study underscores a commitment to rigorous methodology and greatly enhances the reliability of our findings. However, there are limitations to our study that cannot be denied. Firstly, our sample is limited to one city in China, limiting the generalizability of our findings to wider populations. This necessitates further research to ascertain whether our conclusions hold in other contexts. Secondly, our longitudinal analyses were limited to younger participants (median age: 10.33 years). Considering that SUA rise has a threshold with age, our longitudinal results still need to be substantiated by a broader range of studies. Finally, although we have accounted for numerous potential confounding factors, the presence of other unmeasured variables that might have influenced our results cannot be entirely ruled out.

## 5. Conclusions

In conclusion, our study uncovered an intriguing inverted U-shaped, nonlinear association between vitamin D and SUA, as well as the risk of higher SUA status. Furthermore, we demonstrated that higher vitamin D levels had a positive effect on inhibiting SUA increases with age. This study uncovered a novel link between vitamin D and SUA, which holds significant implications for exploring the mechanism of their interaction. Our findings suggest that sufficient vitamin D levels appear to play a preventative role against the age-related increase in SUA. Maintaining optimal vitamin D status may be a strategy for promoting healthy uric acid metabolism.

## Figures and Tables

**Figure 1 nutrients-16-01492-f001:**
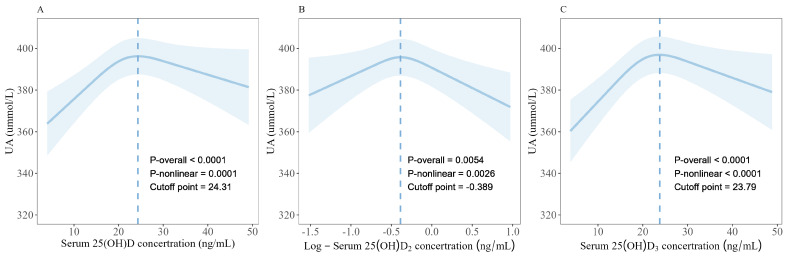
**The dose-response relationship between serum 25(OH)D (A), Log-25(OH)D_2_ (B), 25(OH)D_3_ (C) levels, and serum uric acid.** Restricted cubic splines (RCS) with three knots were used, and the models were adjusted as followed: age, gender, weight status, pubertal development stages, household income, paternal education levels, maternal education levels, smoking and drinking status, vitamin D supplement, multivitamin/mineral supplement, and levels of MVPA. OR, odds ratio; *p*-overall, *p*-values for model tests; *p*-nonlinear, *p*-values for nonlinear tests.

**Figure 2 nutrients-16-01492-f002:**
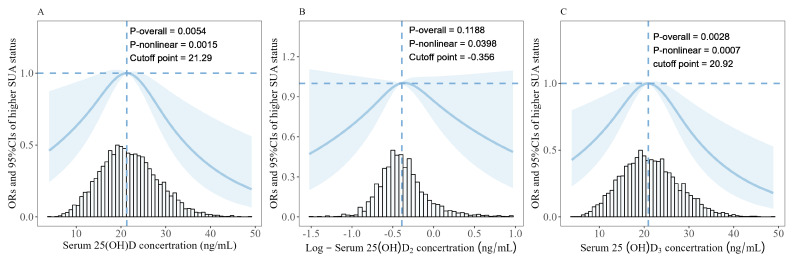
**The association between serum 25(OH)D (A), Log-25(OH)D_2_ (B), and 25(OH)D_3_ (C) le-els and risks of higher serum uric acid status.** Restricted cubic splines (RCS) with three knots, based on the dichotomous logistic regression model, were used, and the models were adjusted as follows: age, gender, weight status, pubertal development stages, household income, paternal education levels, maternal education levels, smoking and drinking status, vitamin D supplement, multivitamin/mineral supplement, and levels of MVPA. OR, odds ratio; *p*-overall, *p*-values for model tests; *p*-nonlinear, *p*-values for nonlinear tests.

**Figure 3 nutrients-16-01492-f003:**
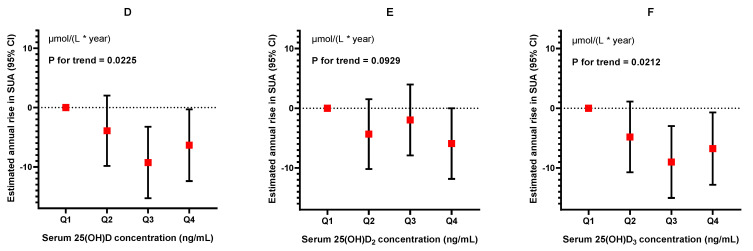
**The association between 25(OH)D (D), 25(OH)D_2_ (E), and 25(OH)D_3_ (F) levels and annual rise in serum uric acid.** Generalized linear models were employed, and the models were adjusted by age, gender, weight status, pubertal development stages, household income, paternal education levels, maternal education levels, smoking and drinking status, vitamin D supplement, multivitamin/mineral supplement, and levels of MVPA. The vertical axis represents the estimate of the annual rise in SUA (β).

**Table 1 nutrients-16-01492-t001:** Baseline characteristics of 4777 participants by the quartiles of serum 25(OH)D.

	Total	Q1	Q2	Q3	Q4	*p*-Value
N	4777	1190	1198	1194	1195	
Age, years	11.94 (9.2–14.7)	14.61 (12.2–16.2)	12.53 (10.2–14.8)	10.91 (8.8–13.4)	9.45 (7.9–12.1)	<0.001
Female, n (%)	2125 (44.5)	701 (58.9)	545 (45.5)	457 (38.3)	422 (35.3)	<0.001
Waist, cm	63.00 (55.80, 70.80)	66.00 (61.00, 72.72)	64.00 (57.70, 71.10)	61.00 (54.00, 69.58)	58.10 (52.50, 66.50)	<0.001
BMI, kg/m^2^	18.07 (15.8–20.8)	19.16 (17.2–21.7)	18.54 (16.2–21.3)	17.52 (15.4–20.4)	16.81 (15.0–19.6)	<0.001
Weight status, n (%)						0.019
Underweight	420 (8.8)	82 (6.9)	99 (8.3)	115 (9.6)	124 (10.4)	
Normal	3270 (68.5)	868 (72.9)	816 (68.1)	1072 (89.8)	789 (66.0)	
Overweight	643 (13.5)	141 (11.8)	167 (13.9)	39 (3.3)	171 (14.3)	
Obesity	444 (9.3)	99 (8.3)	116 (9.7)	83 (7.0)	111 (9.3)	
Puberty stage, n (%)						<0.001
Pre-puberty	1494 (31.3)	113 (9.5)	281 (23.5)	450 (37.7)	650 (54.4)	
Mid-puberty	1060 (22.2)	179 (15.0)	271 (22.6)	340 (28.5)	270 (22.6)	
Post-puberty	2223 (46.5)	898 (75.5)	646 (53.9)	404 (33.8)	275 (23.0)	
Household income, n (%)						<0.001
<CNY 120,000 per year	1254 (26.3)	377 (31.7)	342 (28.5)	270 (22.6)	265 (22.2)	
~CNY 250,000 per year	1603 (33.6)	398 (33.4)	399 (33.3)	416 (34.8)	390 (32.6)	
≥CNY 250,000 per year	1885 (39.5)	404 (33.9)	446 (37.2)	500 (41.9)	535 (44.8)	
Missing	35 (0.7)	11 (0.9)	11 (0.9)	8 (0.7)	5 (0.4)	
Paternal education level, n (%)						<0.001
≤9 years	768 (16.1)	138 (11.5)	151 (12.6)	205 (17.1)	274 (23.0)	
~12 years	1073 (22.5)	223 (18.7)	266 (22.3)	292 (24.4)	292 (24.5)	
~15 years	1333 (27.9)	347 (29.0)	323 (27.1)	336 (28.0)	327 (27.5)	
≥16 years	1556 (32.6)	475 (39.7)	446 (37.4)	352 (29.4)	283 (23.8)	
Missing	47 (1.0)	10 (0.8)	8 (0.7)	13 (1.1)	14 (1.2)	
Maternal education level, n (%)						<0.001
≤9 years	930 (19.5)	168 (14.1)	202 (16.9)	240 (20.0)	320 (26.9)	
~12 years	1241 (26.0)	273 (22.8)	293 (24.5)	328 (27.4)	347 (29.2)	
~15 years	1351 (28.3)	377 (31.5)	365 (30.6)	337 (28.1)	272 (22.9)	
≥16 years	1223 (25.6)	367 (30.7)	329 (27.6)	290 (24.2)	237 (19.9)	
Missing	32 (0.7)	3 (0.3)	5 (0.4)	3 (0.3)	14 (1.2)	
Vitamin D supplement, n (%)						<0.001
No	4078 (85.4)	1089 (91.5)	1032 (86.1)	998 (83.6)	959 (80.3)	
Yes	686 (14.4)	96 (8.1)	165 (13.8)	191 (16.0)	234 (19.6)	
Missing	13 (0.3)	5 (0.4)	1 (0.1)	5 (0.4)	2 (0.2)	
Multivitamin/mineral supplement, n (%)						0.420
No	4321 (90.5)	1095 (92.0)	1077 (89.9)	1069 (89.5)	1080 (90.4)	
Yes	425 (8.9)	87 (7.3)	115 (9.6)	116 (9.7)	107 (9.0)	
Missing	31 (0.6)	8 (0.7)	6 (0.5)	9 (0.8)	8 (0.7)	
Smoking status, n (%)						0.469
Never	4674 (97.8)	1162 (97.6)	1170 (97.7)	1167 (97.7)	1175 (98.3)	
<1 cigarette per month	32 (0.7)	13 (1.1)	8 (0.7)	6 (0.5)	5 (0.4)	
≥1 cigarette per month	21 (0.4)	6 (0.5)	6 (0.5)	6 (0.5)	3 (0.3)	
e-Cigarette	8 (0.2)	2 (0.2)	2 (0.2)	4 (0.3)	0 (0.0)	
Missing	42 (0.9)	7 (0.6)	12 (1.0)	11 (0.9)	12 (1.0)	
Drinking status, n (%)						<0.001
Never	4223 (88.4)	991 (83.3)	1036 (86.5)	1076 (90.1)	1120 (93.7)	
<1 standard drink per month	410 (8.6)	141 (11.8)	128 (10.7)	84 (7.0)	57 (4.8)	
≥1 standard drink per month	120 (2.5)	52 (4.4)	27 (2.3)	26 (2.2)	15 (1.3)	
Missing	24 (0.5)	6 (0.5)	7 (0.6)	8 (0.7)	3 (0.3)	
Levels of MVPA, n (%)						<0.001
<0.5 h per d	1322 (27.7)	426 (35.8)	310 (25.9)	280 (23.5)	306 (25.6)	
~1 h per d	2162 (45.3)	532 (44.7)	534 (44.6)	558 (46.7)	538 (45.0)	
~3 h per d	1089 (22.8)	192 (16.1)	304 (25.4)	296 (24.8)	297 (24.9)	
>3 h per d	161 (3.4)	32 (2.7)	40 (3.3)	47 (3.9)	42 (3.5)	
Missing	43 (0.9)	8 (0.7)	10 (0.8)	13 (1.1)	12 (1.0)	
SUA, μmol/L	343.0 (289.0–415.0)	363.0 (302.0–428.0)	353.0 (297.0–432.0)	337.5 (284.0–412.0)	324.0 (275.0–385.0)	<0.001
25(OH)D, ng/mL	21.44 (17.4–26.0)	14.61 (12.5–16.1)	19.44 (18.4–20.4)	23.64 (22.5–24.7)	29.47 (27.5–32.2)	<0.001
25(OH)D_2_, ng/mL	0.39 (0.27–0.58)	0.37 (0.25–0.56)	0.38 (0.26–0.58)	0.40 (0.29–0.61)	0.41 (0.29–0.59)	<0.001
25(OH)D_3_, ng/mL	20.9 (16.7–25.4)	14.0 (11.9–15.6)	18.9 (17.8–19.9)	23.1 (22.0–24.2)	28.9 (26.9–31.7)	<0.001

Abbreviations: BMI, body mass index; CNY, Chinese yuan; MVPA, moderate-to-vigorous physical activity; SUA, serum uric acid. Continuous variables with normal distributions are reported as means ± SD, while non-normal continuous variables are reported as medians (IQR). Categorical variables are described as frequencies and percentages. We employed analysis of variance (ANOVA) for continuous variables with normal distributions, Kruskal–Wallis rank test for continuous variables with non-normal distributions, and chi-square test for categorical variables to compare the serum 25(OH)D groups.

**Table 2 nutrients-16-01492-t002:** Threshold effect analyses of 25(OH)D, 25(OH)D_2_, and 25(OH)D_3_ on SUA levels using two-piecewise generalized linear regression models.

	Inflection Point (ng/mL)	Concentrations (ng/mL)	N	Model I		Model II	
				β (95% CI), µmol/L	*p*-Value	β (95% CI), µmol/L	*p*-Value
**Serum 25(OH)D ^a^**	24.31	<24.31	3159	12.89 (4.76, 21.01)	0.002	9.15 (1.48, 16.83)	0.019
		≥24.31	1618	−4.85 (−14.09, 4.39)	0.303	−4.41 (−13.38, 4.56)	0.335
**Log-Serum 25(OH)D_2_ ^b^**	−0.389 **^c^**	<−0.389	2497	18.43 (−2.21, 39.07)	0.08	13.17 (−6.37, 32.72)	0.187
		≥−0.389	2280	−22.23 (−35.12, −9.34)	<0.001	−16.80 (−29.22, −4.39)	0.008
**Serum 25(OH)D_3_ ^a^**	23.79	<23.79	3181	1.59 (0.79, 2.40)	<0.001	12.15 (4.56, 19.74)	0.002
		≥23.79	1596	−0.45 (−1.38, 0.47)	0.301	−3.60 (−12.58, 5.38)	0.432

β, estimates of regression coefficients; 95%CI, 95% confidence interval. Two-segment generalized linear models (GLM) were used. Model I includes age and gender. Model II was further adjusted for weight status, pubertal development stages, household income, parental education levels, maternal education levels, smoking and drinking status, vitamin D supplement, multivitamin/mineral supplement, and levels of MVPA. SUA corresponds to units of **^a^** per 10 ng/mL increase; **^b^** per 1 log-serum 25(OH)D_2_ ng/mL increase. **^c^** The value of log-serum 25(OH)D_2_ was −0.389 ng/mL, corresponding to a value of serum 25(OH)D_2_ of 3.32 ng/mL.

**Table 3 nutrients-16-01492-t003:** Two-stage logistic regression analyses the effect of serum 25(OH)D, 25(OH)D_2_, and 25(OH)D_3_ on the risk of higher SUA status.

	Inflection Point (ng/mL)	Concentrations (ng/mL)	N	Model I		Model II	
				OR (95% CI)	*p*-Value	OR (95% CI)	*p*-Value
**Serum 25(OH)D ^a^**	21.29	<21.29	2349	1.52 (1.00–2.35)	0.055	1.45 (0.92–2.31)	0.11
		≥21.29	2428	0.45 (0.30–0.67)	<0.001	0.52 (0.34–0.78)	0.002
**Log-Serum 25(OH)D_2_ ^b^**	−0.356 **^c^**	<−0.356	2797	2.00 (0.92–4.52)	0.086	2.14 (0.93–5.10)	0.081
		≥−0.356	1980	0.54 (0.27–1.03)	0.071	0.71 (0.34–1.39)	0.300
**Serum 25(OH)D_3_ ^a^**	20.92	<20.92	2397	1.86 (1.22–2.86)	0.004	1.68 (1.07–2.66)	0.025
		≥20.92	2380	0.50 (0.33–0.73)	<0.001	0.55 (0.36–0.84)	0.006

β, estimates of regression coefficients; 95%CI, 95% confidence interval. Two-segment logistic models were used. Model I includes age and gender. Model II further adjusts for weight status, pubertal development stages, household income, parental education levels, maternal education levels, smoking and drinking status, vitamin D supplement, multivitamin/mineral supplement, and levels of MVPA. **^a^** Per 10 ng/mL increase; **^b^** per 1 log-serum 25(OH)D_2_ ng/mL increase. **^c^** The value of log-serum 25(OH)D_2_ was −0.356 ng/mL, corresponding to a value of serum 25(OH)D_2_ of 2.27 ng/mL.

**Table 4 nutrients-16-01492-t004:** Longitudinal analysis of the relationship between 25(OH)D, 25(OH)D_2_, 25(OH)D_3_, and a-nual changes in SUA.

	Q1	Q2	Q3	Q4	*p* for Trend
**Serum 25(OH)D**					
N	412	411	408	410	
Model I	0.00 (reference)	−2.97 (−8.89, 2.96)	−7.50 (−13.48, −1.51)	−5.29 (−11.32, 0.74)	0.0519
Model II	0.00 (reference)	−3.91 (−9.84, 2.02)	−9.27 (−15.30, −3.24)	−6.35 (−12.40, −0.30)	0.0225
**Serum 25(OH)D_2_**					
N	412	419	400	410	
Model I	0.00 (reference)	−4.82 (−10.69, 1.05)	−2.48 (−8.43, 3.47)	−6.73 (−12.65, −0.80)	0.0529
Model II	0.00 (reference)	−4.34 (−10.19, 1.51)	−1.96 (−7.90, 3.98)	−5.91 (−11.83, 0)	0.0929
**Serum 25(OH)D_3_**					
N	412	409	410	410	
Model I	0.00 (reference)	−4.34 (−10.27, 1.59)	−7.47 (−13.44, −1.50)	−5.97 (−12.00, 0.06)	0.0431
Model II	0.00 (reference)	−4.80 (−10.74, 1.13)	−9.00 (−15.03, −2.99)	−6.77 (−12.83, −0.71)	0.0212

β, estimates of regression coefficients; 95%CI, 95% confidence interval. Generalized linear models were used. Model I includes age and gender. Model II further adjusts for weight status, pubertal development stages, household income, parental education levels, maternal education levels, smoking and drinking status, vitamin D supplement, multivitamin/mineral supplement, and levels of MVPA. *p*-value for trend was obtained using the median value with each quartile.

## Data Availability

The data described in the manuscript will not be made available due to contractual and privacy restrictions. The codebook and analytic code used for the manuscript are available upon request from wyj@gzhmu.edu.cn.

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
