# Peer review of "The Inverted U-Shaped Association between Serum Vitamin D and Serum Uric Acid Status in Children and Adolescents: A Large Cross-Sectional and Longitudinal Analysis"

_nutrients, 2024, doi:10.3390/nu16101492_

Round 1

Reviewer 1 Report

Comments and Suggestions for Authors

MA Z et al. investigated the association between vitamin D and uric acid (SUA) in a pediatric cohort. They observed an U-shaped relationship between vitamin D and SUA. Sufficient vitamin D levels are protective for the age-related increase in SUA.

 Minor revision:

1.     Correct the number of involved participants in the cross-sectional study in the abstract (line 22). In the text, tables, and figures the number of subjects involved in the cross-sectional study is different from that of abstract  (4777 instead of 4682).

2.     Indicate the age of the participants also in the abstract, otherwise the reader could think that with children you involved participants under the age of 6. (line 22). 

3.     In the abstract it would be appropriate to indicate how the results are reported in the manuscript. If you use standard deviation, odds ratio, and the p-value.

4.     Please provide a definition for the exclusion criteria and the inclusion criteria. Which are the relevant disease-related data you are referring to? (line 108)

5.     Indicate a reference for the standardized methodologies reported in line 137.

Major revision:

1.     Were the serum samples used only for detecting vitamin D and SUA levels? Dosing cytokines may be useful in evaluating the inflammatory state of patients.

2.     Since you stratified patients based on their BMI, do you also evaluate the WHR and WHtR ratio? They could be helpful in describing abdominal fat and providing an idea about the risk of developing cardiovascular diseases. Furthermore, BMISDS could be better in the pediatric age. Please, insert it.

3.     All tables and figures reported in the manuscript lack a comprehensive caption that makes the interpretation of the results understandable to non-expert readers. It would be appropriate for each table and/or figure to indicate the type of observed data, how they are represented (mean ± SD, median, percentages, etc.), and the statistical analysis conducted in the specific case.

4.     Figure 3 seems discordant with that reported in the text and difficult to understand. It could be better to change this graph to one that highlights the association between the serum of vitamin D and the changes in uric acid, also specifying the variables indicated in the axes of the graphs.

5.     In the longitudinal analysis, SUA risk increases with age, especially in subjects presenting low levels of vitamin D (as also reported in line 322; however, from the graphs and the description, this association remains difficult to interpret. It could be better give more data about the age and how this analysis is conducted.

Reviewer 2 Report

Comments and Suggestions for Authors

I congratulate the authors for their interesting work.

- in the material and methods part

As it is also an longitudinal study, please describe with more detail in what consist this part of the study - duration, number of subjects, etc.

- In the results part

row 202 - 203: Compared to the lowest quartile (Q1) group, the groups with higher vitamin D tended to be younger, non-drinkers and had a higher proportion of females. But from the data table it seems that the proportion of women is lower in Q4. Please correct of state more clear your results.

- can you comment more the association with pubertal status and age? can you distinguish between the two variables as they are intricate?

- row 349-350 please rephrase

Comments on the Quality of English Language

Just minor editing errors.
